# HOW BPE AFFECTS MEMORIZATION IN TRANSFORMERS

## ABSTRACT

Training data memorization in NLP can both be beneficial (e.g., closed-book QA) and undesirable (personal data extraction). In any case, successful model training requires a non-trivial amount of memorization to store word spellings, various linguistic idiosyncrasies and common knowledge. However, little is known about what affects the memorization behavior of NLP models, as the field tends to focus on the equally important question of generalization.

In this work, we demonstrate that the size of the subword vocabulary learned by Byte-Pair Encoding (BPE) greatly affects both ability and tendency of standard Transformer models to memorize training data, even when we control for the number of learned parameters. We find that with a large subword vocabulary size, Transformer models fit random mappings more easily and are more vulnerable to membership inference attacks. Similarly, given a prompt, Transformer-based language models with large subword vocabularies reproduce the training data more often. We conjecture this effect is caused by reduction in the sequences' length that happens as the BPE vocabulary grows. Our findings can allow a more informed choice of hyper-parameters, that is better tailored for a particular use-case.

## 1 INTRODUCTION

The Transformer architecture (Vaswani et al., 2017) became the backbone of the state-of-the-art models in a variety of tasks (Liu et al., 2019; Raffel et al., 2019; Adiwardana et al., 2020; Brown et al., 2020). This spurred a significant interest in better understanding inner workings of these models (Vig and Belinkov, 2019; Clark et al., 2019; Kharitonov and Chaabouni, 2020; Hahn, 2020; Movva and Zhao, 2020; Chaabouni et al., 2021; Merrill et al., 2021; Sinha et al., 2021). Most of these works have focussed specifically on how models generalize and capture structure across samples that are similar. For instance, Vig and Belinkov (2019) focussed on how attention align with specific syntactic dependency relations, Hupkes et al. (2020) considered if Transformers generalize *compositionally* and Kharitonov and Chaabouni (2020) studied how different models generalize from very few data. In contrast to these studies, we focus on factors that control the training data memorization behavior of Transformers, which we believe to be important for several reasons.

First, large Transformer models are increasingly often used as a *storage*, for instance, as a general-purpose knowledge base or as a closed-book question-answering system (Petroni et al., 2019; Roberts et al., 2020; Lewis et al., 2020). Clearly, the ability to memorize factual knowledge from the training data is crucial for such applications. There are even Transformers models that are explicitly endowed with an external training data memorization mechanism (Khandelwal et al., 2020; 2021; He et al., 2021), demonstrating that further boosting their memorization abilities is beneficial.

Second, in contrast, the same ability to memorize can become undesirable and lead to a leakage of personal data from trained models (Carlini et al., 2020; Thakkar et al., 2021). A better understanding of the phenomenon is thus instrumental both to enable better memorization when it is needed and to avoid it when not.

Third, while generalization and memorization are often thought of as competing modes of fitting data, training effective models in real tasks requires a non-trivial combination of the two. For instance, successful language models need to generalize to be able to deal with never-seen-before sentences,

but they also need to memorize the spelling of words, the non-compositional meaning of idioms, idiosyncrasies of languages, common knowledge, etc (see, e.g. Dankers et al., 2021).[1]

Despite this apparent importance, there is very little research into memorization in Transformers and in NLP models in general, and we have only superficial understanding of what factors affect this behavior. Intuitively, the number of parameters, data augmentation, and regularization are likely to affect how successful models are in memorization (Zhang et al., 2016; Sablayrolles et al., 2018). In this work, we primarily focus on the influence of a less obvious yet important factor: we study how the selection of modelling units affects memorization. Typically, the same data can be represented on various levels: raw bytes and their groups, individual characters, subword units, and entire words. A very common approach is to learn subword-level vocabulary with Byte-Pair Encoding (BPE) (Sennrich et al., 2015) or similar methods (e.g., Devlin et al., 2018; Kudo and Richardson, 2018; Provilkov et al., 2019). In spite of ubiquitous use of these methods, to the best of our knowledge, there is no clear understanding of how the number of subwords or BPE operations should be chosen and how this affect behavior of a model. We expect that BPE-like segmentation might play a crucial role in memorization, as it controls the trade-off between the number of primitives a model will have to operate with and the lengths of sequences it must represent.

In this work, to characterize a model's behavior, we measure three "facets" of training data memorization. First, as a proxy for the memorization *capacity* of a model, we use its ability to fit random, non-systematic mappings. Next, we study the *preference* for memorization when generalization is possible. For that, we study how easy it is to accurately tell if a particular example was used in the model's training data via a membership inference attack (Shokri et al., 2017). Finally, we examine how easy it is to recover training data from a trained language model. We experiment with three Transformer architectures: causal & masked language models, and encoder-based classifiers.

Our main experimental finding is that, *across all architectures and tasks, the choice of modeling units strongly affects the memorization behavior of the models, with large-cardinality BPE vocabularies greatly facilitating memorization*. This observation holds even when we control for the number of trainable parameters.

After establishing this fact, we look deeper into the causes of the phenomenon we observe. We examine three candidate causes which are principal (side-)effects of applying BPE: (i) removing redundancy in the data (due to compression), (ii) increase in the number of the unique units used to represent the data, or (iii) reducing the length of the training sequences. By finding a similar effect with incompressible randomly generated data we can rule out the first possibility. Next, we artificially double the vocabulary size by introducing "synonym" tokens and observe that the vocabulary growth, in isolation, leads to a different memorization pattern. Thus, by exclusion, we conjecture that reducing utterance length is, at least, a very important factor of memorization.[2].

## 2 STUDYING MEMORIZATION – THE TASKS

To quantify the memorization capabilities and preferences of NLP models, we use three different setups, with which we aim to cover different facets of what one can call training data memorization.

### 2.1 LEARNING MAPPINGS WITH RANDOM LABELS

Firstly, we consider a task of learning non-systematic mappings, where labels are independent from inputs (Zhang et al., 2016). To achieve accuracy above chance, the model has to "store" the training example in some way, thus we assume that a higher training accuracy implies increased training data memorization ability.

To experiment with realistic natural input data, we consider the Stanford Natural Language Inference dataset (SNLI, Bowman et al., 2015). In this dataset, each example is a pair of two sentences, one representing a premise ("A boy is jumping on skateboard in the middle of a red bridge.") and the other representing a hypothesis ("The boy does a skateboarding trick."). Each example is assigned a

---

[1] An interesting example is the language model of Lakhotia et al. (2021), which is trained on sub-phonemic acoustic units without word boundaries, but confidently processes a large vocabulary of English words.

[2] Another potential cause that we consider is the changes in the relative frequencies of tokens that BPE brings along. In Appendix D we investigate and rule out this hypothesis.

label that denotes if the hypothesis entails the premise, contradicts it, or neither contradict nor entails (neutral). We represent examples in a concatenated form with a separator token (::) between a premise and a hypothesis ("A boy is jumping on skateboard in the middle of a red bridge. :: The boy does a skateboarding trick."). For uniformity with other experiments, we transform the dataset into a binary classification task by filtering out all examples with neutral labels.[3] After this filtering, 367,388 examples remain. We replace original labels with randomly sampled ones (-1 / +1, equiprobably), and we measure memorization by measuring the (training) accuracy of the models on this data.

## 2.2 MEMBERSHIP INFERENCE

While the task of memorizing random labels can tell us how much a model can remember, it doesn't allow us to test how much or what a model memorizes when it is trained on tasks which also admit (or require) generalization. As this is typically the case with natural data, we thus need a different strategy to assess memorization in more realistic scenarios. To evaluate memorization in such cases, we resort to measuring models' vulnerability to membership inference attacks (Shokri et al., 2017). Indeed, if it is "easy" to accurately tell if a particular example was used for training, we assume it is actually "stored" in the weights of the model in some form, rather than being inferred from a more general rule or pattern (which would lead to high scores also for examples that were not in the training data, but that are likely given that data).

More formally, suppose we have a model $f_\theta$ that was trained on a subset $\mathcal{D}'$ of a large set of examples $\mathcal{D}, (\mathcal{D}' \subset \mathcal{D})$ with $\mathcal{D}'$ obtained by sampling examples independently from $\mathcal{D}$ with some probability $\lambda$. The goal of membership inference is to figure out, given $f_\theta$, whether a particular example $(x_i, y_i) \in \mathcal{D}$ was included in the training data $\mathcal{D}'$.

We implement a simple membership inference attack protocol by Yeom et al. (2018). Given a model $f_\theta$ parameterized by $\theta$, we calculate its loss on a data point $l(f_\theta(x_i), y_i)$ and compare to a threshold $\tau$: if it is below the threshold, the data point belongs to the training data. By controlling $\tau$ we can control the trade-off between precision and recall of the attack. To avoid the dependency on this parameter and to represent the entire space of possible trade-offs, we use the AUC metric. After training a model, we measure the AUC of the above rule that separates training and hold-out examples.

In this set of experiments, we again use the SNLI dataset. However, in this experiment we use the true labels of the dataset, rather than using the random labels of the previous setup, allowing us to consider both generalization and memorization. To make the prior probability of an example belonging to the training dataset equal to $\frac{1}{2}$, at training time we use only a half of the original's dataset training data (367,388 examples remaining after filtering), with the second half playing the role of the hold-out.

## 2.3 TRAINING DATA RECOVERY

Lastly, we study the memorization capabilities of Transformer models in a setup that is closer to natural large-scale tasks. In particular, we focus on the – interesting for memorization – domain of question-answering, and we consider how well Transformer language models can reproduce exact-match answers to questions present in the training data.

For this experiment, we use the L1 subset of the large-scale Probably Asked Questions (PAQ) dataset (Lewis et al., 2021), which contains 14M queries with candidate answers. As with SNLI, we transform this dataset so that it is suitable for training an LM by concatenating queries and their respective candidate answers, separated by a special token (::). For instance, a training example might be "where is the capital of argentina located :: buenos aires". Whenever PAQ provided more than one answer, we used the first. We lower-cased questions and answers. At test-time, we prompt the trained LM by a query followed with the separator token and check whether the trained LM reproduces the correct answer within top-1 or top-5 results returned by beam search (beam size 5). To speed up the evaluation, we probe a fixed random sample of 4M questions.

---

[3]In the current experiment, this is not strictly necessary, since we map the input examples to random labels. Filtering the data becomes important in the next experiments, in which we instead use true labels.

## 3 MODELS AND HYPERPARAMETERS

We consider three standard Transformer-based architectures: a (causal) language model (LM), a masked language model (MLM), and a sequence encoder (Encoder). In our first two setups (random label memorization and membership inference), we study all three architecture variants. In the experiments on question-answer-recovery, we only study the LM architecture, as those experiments require the ability to sample from the model efficiently.

### 3.1 BPE SETTINGS

The core question that we ask in this paper is how the choice of modeling units affects the memorization behaviour of state-of-the-art Transformer models. To answer this question, for every experiment we create various versions of the involved datasets that differ in the number of subwords, by varying the parameters of the BPE process used to create subwords. In particular, for the SNLI dataset (used in random label memorization and membership inference), we apply BPE with $(0.5, 1, 5, 10, 20) \times 10^3$ steps, resulting in vocabulary sizes of 611, 1097, 4943, 9574, and 18336, respectively. For the larger PAQ dataset, used in the recovery experiment, we get different versions of the dataset by running BPE for $(0.5, 1, 5, 10, 15, 20) \times 10^3$ steps, obtaining vocabulary sizes of 1280, 1784, 5784, 10784, 15784, and 20776, respectively.

### 3.2 CONTROLLING THE NUMBER OF LEARNED PARAMETERS

In our experiments, we vary the size of the subword vocabulary used to represent the data. In turn, the number of the learned parameters that are present in the embedding layer changes, too. It is not unreasonable to expect that the memorization capabilities of a model are impacted by this growth of the number of the learned parameters. To avoid this confounding factor, we complement our study with experiments where we control for the change in the number of embedding parameters. To do so, we replace the embedding layer by a combination of an embedding and a fully-connected layer. This way, we can change the dimensionality of the input & output token embeddings and control the number of the learned parameters while maintaining the rest of the model isolated from any changes. This is a standard architecture variant in `fairseq` (Ott et al., 2019). We report the used embedding sizes and the resulting numbers of parameters in Appendix.

### 3.3 CASTING MODELS AS CLASSIFIERS

Some of the tasks introduced above require that the studied models are binary classifiers (e.g., learning non-systematic mappings § 2.1). Here we discuss how we turn our considered architectures into classifiers. Encoder takes the input sequence of length $l$ and embeds it into a continuous representation $\mathbb{R}^{e \times l}$, where $e$ in the embedding size. We take the embedding of the `eos` token and linearly map it into logits of the labels ($\{-1, +1\}$). Encoder is trained to minimize the cross-entropy error of the target label.

To use LM as a classifier, we associate new tokens (which never occur in the training data) with the target labels and append them to the input strings, after the `eos` tokens. We train the model using the standard teacher-forcing procedure. This way the accuracy of the classifier equates to the accuracy of the predicting last token in a sequence. While this is a non-standard way of training classifiers, it (i) reflects some of the interesting cases where language models are used as universal zero-shot learners (Brown et al., 2020), (ii) allows us to compare LM's memorization capabilities with other architectures directly.

To use MLM as a classifier, we follow a similar approach. We extend an input string by a token that specifies the label. At training-time, this token and a random part of the other tokens are masked. The model is trained to recover all the masked tokens. The task of recovering a masked label resembles how mask-filling is used in knowledge-intensive tasks (Petroni et al., 2019).

### 3.4 MODEL AND TRAINING DETAILS

We use similar Transformer models across experimental setups, with only slight differences, that we describe below. In all experiments, we use the Adam optimizer (Kingma and Ba, 2014). We

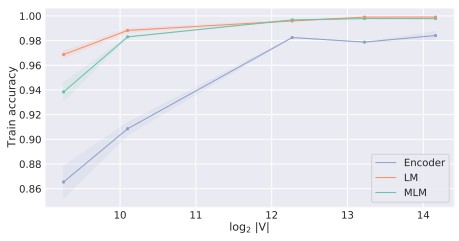 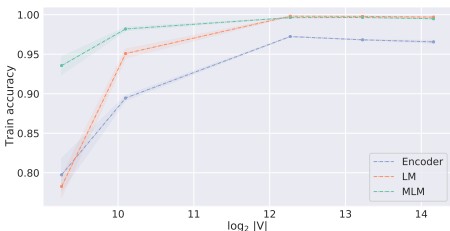

    (a) Vanilla Transformer models.       (b) Models with the number of parameters fixed.

Figure 1: Training accuracy for fitting random labels on SNLI. Shaded area represents $\pm 1$ SEM (Standard Error of the Mean).

use half-precision floats throughout all experiments. We implemented all three architectures using Pytorch (Paszke et al., 2019).

**Memorizing random labels**   The models are trained with learning rate 5e-4. The learning rate is linearly warmed-up for the first 500 updates and then decreased under the square-root schedule. As in this experiment we are interested in the ability to overfit training data, we train the models for 500 epochs. We use batches of 512 examples. If a model achieves training accuracy above or equal to 0.999, the training is stopped earlier. For LM and MLM are implemented as 4-layer, 4-head Transformers with embedding size 256 and FFN layer dimensionality of 512 In preliminary experiments, we found that a 4-layer Encoder model quickly achieves 100% label prediction accuracy (on train), irrespective of the number of subwords. Hence, in this experiment we used a 1-layer version for that model type with embedding and FFN sizes of 256 & 512. When training MLM, each token is masked with probability of 0.2.

**Membership inference**   We use similar architectures and hyperparmeters as above for LM and MLM. Encoder has 4 layers, embedding size of 512 and FFN dimensionality of 1024, same as (M)LM. In this experiment we train for 100 epochs.

**Question Answering**   In this set of experiments, we use the `fairseq` (Ott et al., 2019) implementation of Transformer LM. We study two variants of the architecture: base and large. The base architecture has 6 layers with 8 attention heads, embedding dimensionality of 512 and FFN dimensionality of 2048. The large architecture has 12 layers, 16 attention heads, embdedding dimension of 1024 and FFN dimension of 2048. Both variants have dropout probabilities of 0.1. In this experiment we are only interested in training data memorization, hence we allow all models to overfit and do not apply any early stopping. We stop training after a fixed budget of 70 epochs. We use a learning rate of 5e-4, inverse-sqrt learning rate schedule and a linear warm-up for 4000 updates.

## 4   BPE INFLUENCES TRAINING DATA MEMORIZATION

In this section, we discuss our first main finding: that the number of BPE merges influences the extent to which a model memorizes training data. This finding is persistent across the three different experimental setups we described earlier: random label memorization, membership inference and question-answer recovery. In all these experiments, we systematically control the number of merges, which is roughly the same as the vocabulary size. We also run complementary experiments where we fix the number of learned parameters to stay roughly the same for each vocabulary size (see §3.2).

### 4.1   MEMORIZING RANDOM LABELS

In Figure 1, we report the accuracy on fitting the training data with random labels as a function of base-2 logarithm of the vocabulary size $|V|$. In Figure 1a we report results for "vanilla" architectures (without an additional FFN layer that allows to control for the number of learned parameters) and in Figure 1b we provide results for the models with the number of parameters fixed. We see that for all

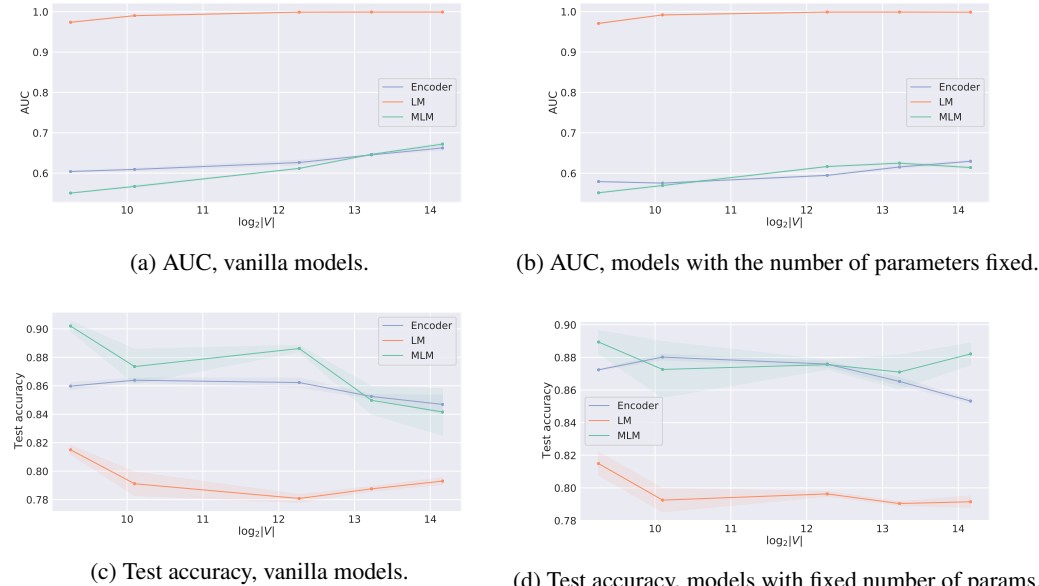

(a) AUC, vanilla models.

(b) AUC, models with the number of parameters fixed.

(c) Test accuracy, vanilla models.

(d) Test accuracy, models with fixed number of params.

Figure 2: Membership inference attack, SNLI dataset. Shaded area represents $\pm 1$ SEM.

three models, LM, MLM, and Encoder, the dependency is very consistent: as the vocabulary size grows, the models become universally more successful in fitting random labels. Encoder fits around 87% of the random labels with the subword vocabulary size of 611 and has a training accuracy of more than 98% when the vocabulary size is increased to 18,336. LM and MLM – that have more layers than Encoder – start off with a slightly higher accuracy (94% and 97%, respectively), their accuracy quickly climbs up to 100% when increasing the number of subwords. This tendency persists when controlling for the number of learned parameters (Figure 1b).

## 4.2 MEMBERSHIP INFERENCE

In Figure 2 we report the AUC for our membership inference experiments for all three architectures (recall that these experiments used the same SNLI data as the previous experiment, except with with true instead of random labels). Mirroring the results of our previous experiment, we observe a monotonic dependency of the membership attack success rate in function of the size of the vocabulary used: for all architectures, larger vocabulary granularity implies more memorisation.

In Figures 2c & 2d, we report the accuracy on the same exact models on the hold-out data. We see that all models achieve decent generalization, with accuracy above 0.78. From Figure 2c we see that as the vocabulary size grows, the test accuracy of (M)LM has distinct regions of growth. We believe this indicates two important points: (i) generalization is not directly at odds with memorization, and (ii) there is a level of granularity that allows better memorization *and* better generalization.

## 4.3 QUESTION ANSWER RECOVERY

In Figure 3 we report how the top-1 (Figure 3a) and top-5 (Figure 3b) accuracies change in function of the logarithm of the vocabulary size. We report results for Transformer-base and Transformer-large modifications, alongside with the parameter-controlled variants. We firstly observe that the model size has a large impact on memorization: LM-large outperforms LM-base in all cases. Next, we see that both models confirm the pattern we observed in the previous experiments: vocabulary growth consistently leads to a growth in the memorization accuracy.

Focusing on the models with the number of parameters fixed, we see that even when correcting for the total number of embedding parameters, the size of the learned vocabulary greatly affects the ability of recovering training data.

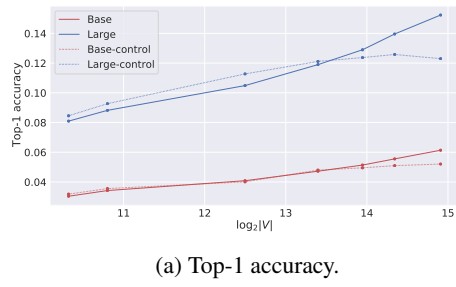 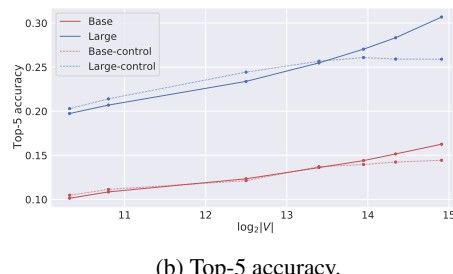

(a) Top-1 accuracy.                                    (b) Top-5 accuracy.

Figure 3: Training data recovery: top-1 and top-5 accuracy on extracting the correct answer when prompted with a query.

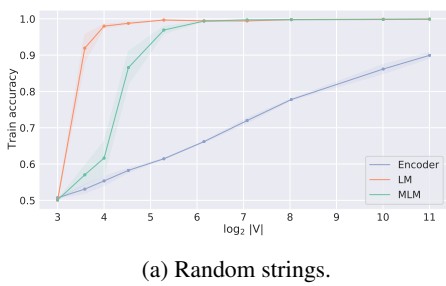 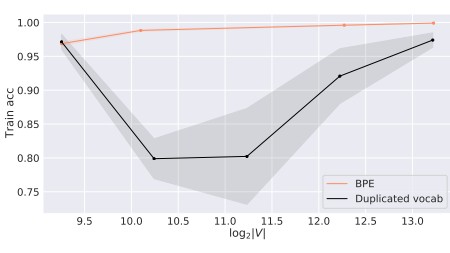

(a) Random strings.                        (b) SNLI. BPE vs. Duplicated vocab.

Figure 4: Analysing potential causes of the increased memorization: fitting random labels on random strings (left) and SNLI (right) datasets. Everywhere shaded area represents $\pm 1$ SEM.

## 5 LOOKING FOR EXPLANATIONS

In §4 we established that larger vocabularies, learned by BPE, lead to stronger memorization in Transformer models, even when we control for the number of the learned parameters. In the current section, we focus on why this might be the case, and further investigate this observed effect.

We hypothesise that the primary cause of the observed behavior is that with the growth of the BPE vocabulary the input sequences become shorter (*length hypothesis*). In turn, shorter sequences are easier to memorize as the work of attention is simplified. Indeed, it is known that learning complex attention patterns is hard and often multiple attention heads turn out to be useless, which potentially limits the ability to memorize complex "explanations" of the data (Voita et al., 2019; Michel et al., 2019). At the same time, shorter sequences would shift the responsibility for memorization onto FFN layers, which are known to memorize well (Zhang et al., 2016).

However, there could be two alternative explanations at play. Being a compression algorithm originally, BPE compresses the data and it could be easier to memorize data without redundancies (*redundancy hypothesis*). Second, BPE increases the vocabulary size and it is possible that each individual token becomes more predictive of the label (*vocabulary size hypothesis*). As an illustration, in the limit case of the vocabulary growth, every sequence might have a unique token. [4] In this case, there is a one-to-one mapping between those tokens and labels (answers) that will, again, simplify the work of self-attention. In a series of experiments below, we contrast out these three hypotheses.

### 5.1 EFFECT PERSISTS ON INCOMPRESSIBLE DATA

To investigate the redundancy hypothesis, we experiment with learning on randomly generated incompressible data and examine whether a similar behavior persists.[5] We generate random data by enumerating all $2^l$ binary ($V = \{0, 1\}$) sequences of length $l$ and randomly assign each sequence to

---

[4]Generally, BPE stops before that, as it is does not cross word boundaries.

[5]Note that in §4.1 we experimented with random *labels*; here, the input sequences are random, too.

one of two classes, $\{-1, +1\}$. To study how the chosen level of representation affects the models' memorization capacity, we apply BPE with $\{0, 4, 8, 16, 32, 64, 128, 256\}$ steps on the generated data. This allows us to start from a binary dataset with sufficiently large sequences and produce a sequence of datasets *of the same storage complexity* (in bits), but with varied vocabulary sizes.

In Figure 4 we report the accuracy on fitting the training data with random labels as a function of the logarithm of the vocabulary size. Again, we observe that for all three architectures the dependency is very consistent: as the vocabulary size grows, the models become universally more successful in fitting random labels.[6] We conclude that the increased memorization success is thus not caused by BPE compressing-out redundancies in more natural data.

## 5.2 VOCABULARY SIZE GROWTH DOES NOT EXPLAIN BETTER MEMORIZATION

To investigate the *vocabulary size* hypothesis we set up an experiment in which we increase the vocabulary size, while keeping sequence lengths constant. We start from our earlier random label memorization setup with SNLI data, using the vocabulary learned with 500 BPE operations. To increase the number of tokens in this vocabulary, we randomly replace each unique token's occurrence with either of two new tokens, with equal probability, roughly doubling the vocabulary size. [7] For instance, say that the current vocabulary has a token "cat". We then iterate over the corpus and replace half of its occurrences with "cat1" and the other half with "cat2" (making sure that those do not happen in the prior vocabulary). Thus, if the vocabulary growth alone can explain the observed effect, we will see that the datasets with "duplicated" tokens would also be easier to memorize.

We report results of this experiment using the LM model in Figure 4b, contrasting it to the BPE-induced memorization results. We see that the resulting curves exhibit considerably different patterns. The line that corresponds to the doubling procedure has a sharp fall in the beginning. Later, as the vocabulary size grows, in accordance with in our thought experiment, tokens become increasingly more unique to sequences which boosts the train accuracy. In contrast, the line obtained by growing the vocabulary via BPE shows a monotonic, consistent growth for all vocabulary sizes. From that we can conclude that the observed phenomenon cannot be explained solely by the vocabulary growth.

To summarize, the above experiments allowed us to rule out the alternative explanations (redundancy and vocabulary growth hypotheses) and *we conclude that reduction of the sequence length is the primary factor of the observed memorization effect.*[8]

## 6 DISCUSSION

While the generalization abilities of state-of-the-art models in NLP have been quite extensively studied in the recent past (Finegan-Dollak et al., 2018; Hupkes et al., 2018; Lake and Baroni, 2018; Keysers et al., 2019; Korrel et al., 2019; Mul and Zuidema, 2019; Raunak et al., 2019; Saxton et al., 2019; Kim and Linzen, 2020; Dankers et al., 2021; Vig and Belinkov, 2019; Dubois et al., 2020; Hupkes et al., 2020; Kharitonov and Chaabouni, 2020) comparatively little is known about what factors impact how such models *memorize*.

For modelling natural data, however, both generalization and memorization are relevant properties. In some cases, because memorization is harmful – for instance when personal data leak from a trained model. In other cases, memorization instead is necessary for accurate performance – for instance to recall the spelling of words or the meanings of idioms. In this work, we therefore focus specifically on memorization, considering in particular the impact of the choice of modelling unit, which recently became particularly relevant given that all current SOTA NLP architectures are trained with an experimentally tuned number of *subwords*.

We studied memorization behavior of three types of Transformer models (causal and masked language models, and an Encoder-based classifier). We looked at memorization from three different perspectives: the ability to memorize random mappings, vulnerability to membership inference

---

[6]In this experiment, for small vocabulary sizes, the total number of learned parameters is nearly constant.

[7]For instance, nothing will change for a token that occurs once in the dataset.

[8]A flip side of this conjecture is that a more expressive attention improves memorization. In Appendix we show this to be the case: increasing the number of attention heads improve random sequence memorization.

attacks, and ability to directly reproduce training data on a prompt. We have observed a strong and monotonic influence of the BPE vocabulary size on how successful Transformer models are at memorizing training data. With higher granularity vocabulary (i.e., more BPE merge operations), Transformers (a) *can* memorize random mappings better, hence they have higher memory capacity, and (b) *do memorize* more of training data in realistic tasks.

We considered several explanations for this strong trend. Is the increase perhaps *due to the increase in the number of trainable parameters*? Often, the embedding layer has a noticeable share of the trained parameters and this number grows with the size of the vocabulary. We show that while this growth does play a role, the effect persists when we ensure that the number of trained parameters remains constant when the granularity of the vocabulary grows.

Next, we conjecture that the phenomenon is caused by the reduction in the sequence length that correlates with using larger-cardinality BPE vocabularies. It can simplify memorization, as the latter will no longer require learning complex attention patterns "explaining" the data and offset complexity onto fully-connected layers, which are known to be good in memorization (Zhang et al., 2016). In contrast, learning useful attention patterns is hard and often multiple heads turn out to be useless (Voita et al., 2019; Michel et al., 2019). Hence, it is possible that self-attention serves as a bottleneck that limits the memorization capacity.

There are, however, two alternative explanations of the observed trend. It can happen that compressed data with less redundancy is easier to memorize (*redundancy* hypothesis). However, our experiment on random artificial data (Section 4.1) indicates that the reported effect holds even when the data is not compressible in the information-theoretic sense. Further, can *the growth of the vocabulary size* explain the behavior? It does not seem so: our experiment with artificially duplicating tokens shows that this only starts to improve memorization with relatively large vocabularies and is detrimental for memory at the beginning. After ruling out the two alternative possibilities, we are left with our initial hypothesis that using higher-cardinality BPE subword vocabularies implies having to manage shorter sentences on average, which in turn enable easier memorization.

With our work, we contribute to the important question of what impacts memorization behaviour in Transformer models. We believe that our findings, firstly, have an important practical implication. Our experiments provide a clear demonstration that both in artificial and natural data, the simple choice of the number of subwords used has a large impact on the extent to which models memorize training data, a fact that is not at all obvious when merely looking at performance on i.i.d. train/test splits. This therefore calls for a more careful consideration of how many subwords to use for a particular task, a decision that, in practical scenarios, is rarely thoroughly motivated or investigated. In cases where a lot of memorization is desirable (e.g., QA-tasks) or undesirable (e.g., public models trained on medical data), the number of subwords could provide an important factor in tuning the amount of memorization that happens inside a model. Our findings can provide guidance for tuning the learned data structures (e.g., learned Bloom filter-like existence indexes (Kraska et al., 2017)) and compression algorithms that train a Transformer model on the data-to-be-compressed as a compression step (Izacard et al., 2019; Bellard, 2021). If increasing the subword vocabulary size is not possible, boosting expressiveness of the attention can provide another solution (Appendix A).

Second, our work provides an interesting perspective about the relationship between memorization and generalization. As we already mentioned before, memorization and generalization are often thought of as competing modes of fitting data. Even in work that explicitly acknowledges that both are required to model natural language (e.g. Dankers et al., 2021), they are still seen as skills that are required in different situations: some examples, or phrases require memorization, while others require generalization. However, our experiments with SNLI data show the relationship between memorization and generalization are not directly at odds with each other, there relationship is more complex than that: there appears to be a level of granularity that allows better memorization *and* generalization. This experiment therefore begs the question: what is the actual relationship between memorization and generalization in neural networks? To what extent does one help the other? Is this related to the age-old *processing vs storage* question in human language processing (Jackendoff, 2007; Pinker, 1999; Pinker and Prince, 1988; Clahsen, 1999; Rumelhart and McClelland, 1986)?

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

## A  NUMBER OF HEADS

In our analysis in §5, we conjecture that the observed effect is caused by the fact that large-cardinality BPE vocabulary reduces pressure on attention to learn complex patterns due to having less tokens to attend to. This raises the next question: if we increase the power of attention (e.g., by growing number of heads) would the memorization be improved? To investigate that, we repeated our random-string memorization experiment. This time, we have fixed the BPE vocabulary size (8 merges) and varied the number of heads used in each layer as $\{1, 2, 4, 8, 16\}$ and measured the training accuracy.

Our results in Figure 5 confirm our expectations: more heads allow more successful memorization. Indeed it seems that the attention serves as a representation bottleneck that does not allow to easily model complex interactions between tokens and, in turn, limits the memorization. What we observe in this paper is that, when needed, this limitation can be alleviated by (a) increasing granularity of the units that represent the data (e.g., by running BPE), or (b) improving representation power of the self-attention mechanism.

## B  HYPERPARAMETERS USED IN EXPERIMENTS

### B.1  RANDOM LABELS, SNLI

In all experiments we use fixed (sinusoidal) positional embeddings. We use starting learning rate of 5e-4 that is linearly warmed-up for 500 epochs and then decayed under the inverse-sqrt rule. We use half-precision floats to represent the weights. Batch size is 512. In these experiments, we disabled dropout. LM & MLM models have 4 layers while Encoder has 1. For each BPE vocabulary size, we repeated our experiments with 3 different random seeds. The attention layers have 4 heads. Hidden and embedding layers have dimensionalities of 512 and 256. Depending on the experiment, we used 1 or 4 GPUs to train each model, maintaining the overall batch size constant.

### B.2  MEMBERSHIP INFERENCE

For (M) LM we used the same parameters as in § B.1. In this experiment, Encoder has 4 layers, embedding size of 512 and FFN size of 1024. We trained for 100 epochs.

### B.3  QA

We used the standard Transformer LM models of `fairseq` that are specified by the `--arch=transformer_lm_big` (large) and `--arch=transformer_lm` (base).

We use fixed (sinusoidal) positional embeddings. The base architecture has 6 layers with 8 attention heads, embedding dimensionality of 512 and FFN dimensionality of 2048. The large architecture has 12 layers, 16 attention heads, embdedding dimension of 1024 and FFN dimension of 2048. Both

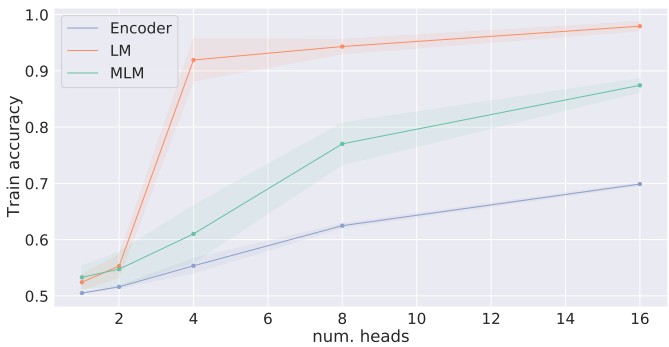

Figure 5: Random label memorization accuracy vs. number of heads.

| BPE steps | token embedding size | num. parameters |
|---|---|---|
| 500 | 4400 | 18,311,749 |
| 1K | 3050 | 18,239,335 |
| 5K | 900 | 18,238,681 |
| 10K | 480 | 18,105,192 |
| 20K | 260 | 18,231,474 |

Table 1: LM: input embedding size and number of learned parameters for the control models.

| BPE steps | token embedding size | num. parameters |
|---|---|---|
| 500 | 4400 | 18,316,149 |
| 1K | 3050 | 18,242,385 |
| 5K | 900 | 18,239,581 |
| 10K | 480 | 18,105,672 |
| 20K | 260 | 18,231,734 |

Table 2: MLM: input embedding size and number of learned parameters for the control models.

variants have dropout probabilities of 0.1. All models are trained for 70 epochs. We use a learning rate of 5e-4, inverse-sqrt learning rate schedule and a linear warm-up for 4000 updates. Again, we used half-precision floats to represent models' weights. The input and output embedding matrices are shared in the models. Each GPU's batch contains up to 3072 tokens and accumulate gradients from 8 batches before running an update. The models were trained on 8 GPUs each.

## B.4 Random sequences

As in § B.1.

## B.5 Duplicated tokens experiment

As in § B.1.

## B.6 Control models

In experiments where we control the number of learned parameters, input token embedding size is no longer attached to the embedding size within the Transformer model (which remains identical to the vanilla, non-ctontrolled model). In Tables 1-6 we report the embedding sizes and the number of learned parameters for control models we used in experiments.

## C Training Data Extraction with Sampling

In this Section we investigate whether the relation between the success rate in the training data extraction experiments and the BPE-learned vocabulary size, reported in Section 4.3 and Figure 3, is independent from the way we generate continuations of a prompt.

We re-evaluate the training data extraction performance for the large models used in Section 4.3. We follow exactly the same protocol as in Section 4.3 with one exception: instead of using beam search, we sequentially sample tokens from the language model (i.e., we run ancestral sampling) at temperature 1.0. For each prompt, we sample 20 candidates and measure how often the ground-truth continuation is among those. We report our findings in Figure 6. From Figure 6 we see that the reported relation has the same form as before: larger BPE vocabulary sizes lead to better memorization. Overall, we conclude that our findings are likely to be independent from the specific training data extraction method used.

| BPE steps | token embedding size | num. parameters |
|-----------|---------------------|-----------------|
| 500 | 1125 | 1,792,731 |
| 1K | 785 | 1,791,741 |
| 5K | 231 | 1,787,673 |
| 10K | 125 | 1,788,106 |
| 20K | 67 | 1,790,056 |

Table 3: Encoder: input embedding size and number of learned parameters for the control models (1 layer).

| BPE steps | token embedding size | num. parameters |
|-----------|---------------------|-----------------|
| 500 | 3000 | 13,322,138 |
| 1K | 2300 | 13,294,038 |
| 5K | 820 | 13,305,718 |
| 10K | 460 | 13,287,138 |
| 20K | 220 | 12,670,778 |

Table 4: Encoder: input embedding size and number of learned parameters for the control models (4 layer).

## D  CAN CHANGES IN FREQUENCY DISTRIBUTION CAUSE THE OBSERVED EFFECT?

In the main text, we considered three factors that accompany growing BPE vocabulary sizes: reduction in sequence length, increased number of tokens, and decrease in the redundancy. Another potential confounding behavior is that BPE affects the (relative) frequency of the tokens used for modelling the data.

To showcase this, we run the following experiment. We take BPE vocabularies learned for the SNLI dataset with the number of merges varied in $\{100, 400, 1000, 5000\}$ and, for each token, measure its frequency in the corpus. Next, we sort the tokens in the order of decreasing frequency and plot token frequency in function of its order after sorting. We can expect that with the increased number of merges, BPE-learned subwords will start approximating words more closely and the frequency distribution of tokens will approach that of words. In turn, the latter follows an extremely skewed Zipf distribution (Zipf, 1949). Figure 7a supports this prediction: increasing the number of merges forces the token distribution to become more peaky.

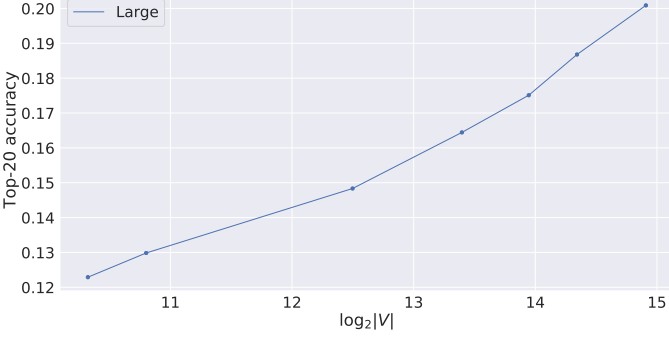

Figure 6: Training data recovery: top-20 accuracy on extracting the correct answer when prompted with a query (sampling).

| BPE steps | token embedding size | num. parameters |
|-----------|----------------------|-----------------|
| 500 | 3000 | 162,139,136 |
| 1K | 2900 | 162,269,536 |
| 5K | 1420 | 162,278,176 |
| 10K | 860 | 162,192,256 |
| 15K | 620 | 162,212,576 |
| 20K | 480 | 162,112,256 |
| 30K | 335 | 162,150,096 |

Table 5: LM-Large, QA experiment: input embedding size and number of learned parameters for the control models.

| BPE steps | token embedding size | num. parameters |
|-----------|----------------------|-----------------|
| 500 | 2400 | 24,444,928 |
| 1K | 1965 | 24,433,048 |
| 5K | 811 | 24,443,424 |
| 10K | 468 | 24,441,472 |
| 15K | 328 | 24,428,352 |
| 20K | 253 | 24,430,728 |
| 30K | 174 | 24,447,136 |

Table 6: LM-Base, QA experiment: input embedding size and number of learned parameters for the control models.

*Can the increased skeweness of the token distribution explain the increased memorization?* To rule this possibility out, we run an experiment where the token distribution is fixed to be uniform, but the sequence length is reduced.

To achieve this, we start from the same dataset of random strings as in Section 5.1. In this dataset, the initial distribution of 0s and 1s is uniform as we enumerate *all* binary strings of a fixed length. Next, we replace BPE with an "idealized" subword learning procedure which maintains the uniform distribution of tokens but reduces the lengths of the strings. The procedure is simple: we firstly group all possible pairs of 0s and 1s into new tokens (00, 01, 10, 11) and use them to encode the initial strings (e.g., "0 0 0 0 0 0 0 0 0 0 0 0 0 0 0 1" becomes "00 00 00 00 00 00 00 01"). We recursively repeat this procedure, obtaining a sequence of views of the same dataset, at each step growing the vocabulary size quadratically (2, 4, 16, and 256) and reducing the string length by two. In this process, all tokens remain uniformly distributed.

Using the obtained series of datasets, we repeat the random label memorization experiment (Section 5.1) and report the results obtained in Figure 7b. We observe that the reported effect persists even in the case when the relative frequencies of tokens are not changed and fixed to be uniform, thus disproving the hypothesis that the observed effect is due to frequency and not token length.

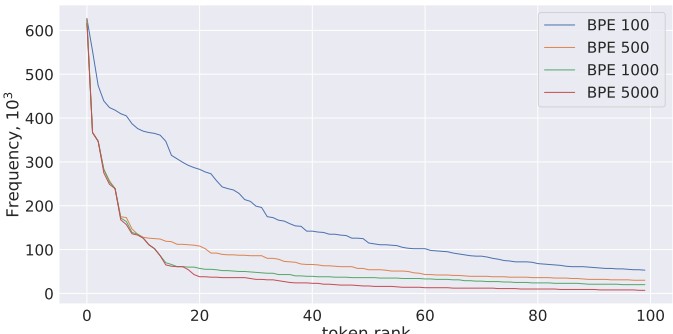

(a) Token frequency in function of its frequency rank (position in the list of tokens when sorted in the decreasing frequency order) for BPE vocabularies of different size. SNLI dataset.

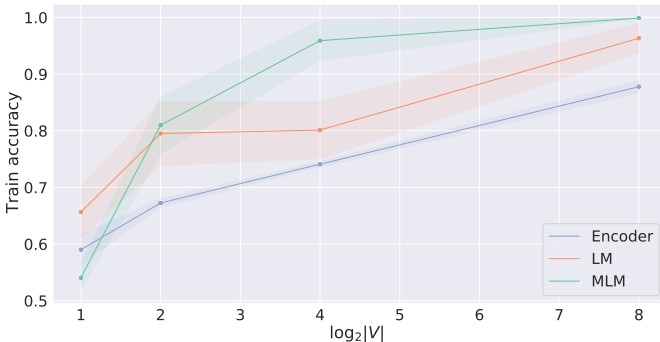

(b) Accuracy on fitting random labels on random strings. The vocabulary size is controlled by a procedure that maintains uniform distribution of the tokens (see the text).

Figure 7: Investigating the relative frequency hypothesis.

