# OpenReview forum: "How BPE Affects Memorization in Transformers"
_ICLR.cc/2022/Conference — ICLR 2022 Submitted_

### Official Review · Reviewer_WXGB · 2021-11-01

**Correctness:** 3
**Technical Novelty And Significance:** 3
**Empirical Novelty And Significance:** 3
**Recommendation:** 6
**Confidence:** 5

**Main Review:**

Pros:
1. The experiments are comprehensive. The authors design three different types of experiments to thoroughly compare the memorization ability differences between three different models (Encoder, LM, MLM).
2. The paper explores the relationship between the memorization and generalization capabilities of the model, and shows that generalization is not directly at odds with memorization, which can inspire future model design.

Cons:
1. The authors focus on three candidate causes, which the authors claimed are principal (side-)effects of applying BPE: (i) removing redundancy in the data (due to compression), (ii) increase in the number of the unique units used to represent the data, or (iii) reducing the length of the training sequences. Other factors that are closely related to the BPE vocabulary may be also important, such as the subword frequency. Can the authors explain why the three factors examined in this paper are more likely the causes than other factors (e.g., subword frequency).
2. Although the paper presents several interesting conclusions, they can not be directly used to determine a suitable vocabulary size in practical tasks, where the demand of the model's memorization is unknowable.


Concerns:
1. The conclusions in this study are only validated on the SNLI dataset (PAQ is only used for the training data recovery experiment), which may threat the universality of the findings. Experiments on other datasets and tasks (e.g. the machine translation task where Transformer was invented) are necessary.

**Summary Of The Paper:**

This paper investigates the impact of subword vocabulary size on the memorization ability of Transformer models. The authors designed three types of tasks to evaluate the changes of model memorization, namely learning mappings with random labels, membership inference and training data recovery. Experimental results show that the memorization ability of Transformer model is stronger with the increase of subword vocabulary size, which the authors attribute to the reduction in the sequences' length.


**Summary Of The Review:**

The research questions explored in the work are very interesting and important. However, the experiments are relatively weak, and it is not clear how to apply their findings in practical tasks.

---

> ### Author Response · Authors · 2021-11-19
> **Authors' reply to Reviewer WXGB**
>
> We thank Reviewer WXGB for their feedback. Below we reply to their main points:
>
> > Other factors that are closely related to the BPE vocabulary may be also important, such as the subword frequency.
>
> Following your comment and a comment by Reviewer saPE we looked into whether changes in the relative frequency of the tokens due to BPE might cause the observed memorization phenomenon. Our study is reported in the Appendix D of the updated text. Below is a summary of it.
>
> We start by showing that, indeed, the size of the BPE-learned vocabulary affects the relative distribution of the tokens, with larger vocabulary size making it increasingly peaky. Next, we take a dataset of randomly generated strings (same as in the main text S5.1) that has uniform distribution of tokens (0s and 1s) and repeatedly apply an "idealized" subword learning procedure that groups all possible pairs of tokens. At each step, this procedure maintains the uniform distribution of tokens, shortens the string length by a factor of two and grows the vocabulary size quadratically. Using the obtained sequence of dataset "views" we repeat the random label memorization experiments and confirm that, still,memorization becomes increasingly successful as string length (measured in tokens) decreases. Thus we can also rule out the possibility that the observed effect is due to changes in the relative frequencies of tokens - as it persists when there is no change in frequencies.
>
>
> > Can the authors explain why the three factors examined in this paper are more likely the causes than other factors (e.g., subword frequency).
>
> We believe that we have reliably established the core fact that BPE vocabulary size greatly affects the ability and the tendency of the Transformer-based architectures to memorize training data. Looking into it further, we have ruled out three of four most likely explanations (reduction in string length, change in number of tokens, redundancy removal and relative frequency of tokens). While it is hard to prove that there are no other potential confounding factors, we hope that our experiments have significantly narrowed the space of explanations and strongly point towards the conclusion that sequence length plays a central role here.
>
> > Although the paper presents several interesting conclusions, they can not be directly used to determine a suitable vocabulary size in practical tasks, where the demand of the model's memorization is unknowable.
>
> Thanks for your comment. We agree that in the most standard scenarios where a practitioner solely optimizes for performance on hold-out data, the required level of memorization is not known beforehand.
> However, we believe that it is very useful to know which hyperparameters actually affect memorisation. Then, in situations where it is important, a researcher can have a clear  intuition on what parameters to look at. For instance, if a model is destined for a public release, with everything else being equal, a researcher/developer might prefer a model trained on representations that are less susceptible to memorization. This would augment the existing arsenal of privacy preserving approaches, such as differential privacy.
> As another example, our insights into the factors of memorization can be useful when designing compression tools based on Transformer-based ML models [1], where memorization of the training data is beneficial.
>
> [1]  Fabrice Bellard, Lossless Data Compression with Neural Networks, https: //bellard.org/nncp/nncp.pdf.
>
> > The conclusions in this study are only validated on the SNLI dataset (PAQ is only used for the training data recovery experiment), which may threat the universality of the findings. Experiments on other datasets and tasks (e.g. the machine translation task where Transformer was invented) are necessary.
>
> Thanks for the suggestion. We are trying to get a series of MT-based experiments before the end of the rebuttal period, and we will promptly update you.

---

> ### Author Response · Authors · 2021-11-29
> **Reply to Reviewer WXGB (2)**
>
> >  Experiments on other datasets and tasks (e.g. the machine translation task where Transformer was invented) are necessary.
>
>
> Thanks again for your suggestion to check if our findings generalize to an NMT task.
>
>
> Unfortunately, we seem to be unable to update the draft directly. Hence we anonymously share our additional experiment using GoogleDrive: https://drive.google.com/file/d/1gA29SGTGa8hMFUeBLHgPnh8PLPDu44S9/view which we will add in Appendix of the paper. In this pdf, we provide a preliminary study on how BPE vocabulary size affects effectiveness of membership attacks on NMT models. In line with our earlier findings, our new results suggest that membership inference attacks become increasingly successful as BPE vocabulary grows.
>
> Here is a short summary of the experiment. We based it on an English-Dutch translation task, using the OPUS dataset. We randomly sampled two equally-sized 8M sentence sets as training and hold-out sets and used FLORES-101 dev-set for early stopping. Using English and Dutch BPE vocabularies with 10k, 20k, 40k, and 80k merges we trained Transformer-base seq2seq NMT models following a standard fairseq NMT recipe. Finally, we evaluated AUC of the membership inference attack in a procedure similar to that described in S4.2. As a result, we see that models trained under higher-cardinality vocabularies are universally more susceptible to membership inference attacks. Thus we conclude that our findings also apply to Transformer seq2seq models used in NMT.

---

> > ### Comment · Reviewer_WXGB · 2021-12-06
> > **Thanks for the new experiments on machine translation**
> >
> > Thank you for the additional experiments on machine translation tasks, which make the findings more convincing. I will increase my score for the new experiments and detailed response, which I hope can be included in the final version.

---

### Official Review · Reviewer_saPE · 2021-11-02

**Correctness:** 3
**Technical Novelty And Significance:** 2
**Empirical Novelty And Significance:** 3
**Recommendation:** 6
**Confidence:** 4

**Main Review:**

This work is placed on the border for me as it is investigating well crafted question and does some empirical validation of multiple hypotheses. On the other hand, there is not much of discussion on what can we do with long sequences when we require memorization? Authors provide very high-level paths such as 'Our findings can provide guidance for tuning the learned data structures', but I would be very curious to read more details. I think it would nicely fit as findings paper rather than work with the novel contribution.

When it comes to strengths, I like the construction of memorization properties which allows to quantify the memorization by using the proposed probing tasks. In addition, authors tried to provide many details on specific hyper-parameters they used, although I left some comments below where more details would be welcome.

Regarding weaknesses: while the memorization properties are clearly outlined, the construction of model parameterization in probing tasks is written on very high-level without any discussion of why this specific method of casting the model as classifier is chosen and how it may or may not provide necessary information towards research questions about memorization, this is mainly addressing section 3.3).
In addition, authors mention 'SOTA \ state of the art' several times during the work while they do not much relate to the actual SOTA models. Please correct me here if I am wrong, but considered models are not in the same ballpark as SOTA models. I want to stress that the latter fact is not a bad thing, but then there is no need to relate with SOTA then.

Below I have noted multiple comments where it was hard to me to get all the details or where I felt the minor change / addon would improve the presentation:

Page 3, *"At test-time, we prompt the trained LM by a query followed with the separator token and check whether the trained LM reproduces the correct answer within top-1 or top-5 results returned by beam search (beam size 5)."* : This is very specific design choice, why did you choose it? Model conditioned on the q gives the entire distribution on sequence-level and beam search is very sensitive to the beam size? I see an alternative as doing unbiased sampling and computing expected similarity with the answer. Some discussion on that would help to clarify this.

Bottom of page 3: you used word *setups* to refer to both tasks and models (right?) which is a bit confusing.

Section 3.3: as I wrote above, in my opinion this section needs to be larger as it defines very important design of the model parameterization, right now it is not even clear enough what are 'target class' in the context of vocabulary tokens? I believe explaining this would improve the presentation of that part a lot.

Page 5: *achieves 100% train accuracy* : in cases of MLM, was acc measured only on task class prediction or on other masked tokens as well?

Page 5: *Hence, in this experiment we used a 1-layer*:  Could you explain why you chose to reduce the model size? In my understanding the task itself might be too easy so the other way is to alter the underlying task (since you specifically construct it).

Page 5: *which was enough for the training loss to converge*: Converge to what? Was there an early stopping criteria on loss or acc?

Page 8: *Thus, if the vocabulary growth alone can explain the observed effect, we will see that the datasets with “duplicated” tokens would also be easier to memorize.* : In your proposed construction the frequency distribution of tokens doesn't change relatively to each other, right?. Is it the same for BPE learning when number of merges is growing? Would be great to show this, otherwise it is hard to buy this statement.

**Summary Of The Paper:**

This paper studies the memorization properties of an NLP models conditioned on how large is the vocabulary size. They concentrate on widely used BPE algorithm in order to split original data into subword units. Further they construct a test-bed consisting on several tasks where each task is related with a specific memorization aspect connected to the model such as capacity or preference which authors introduced themself.

Experiments empirically validate the change of memorization properties as the vocabulary size is changing where it exhibits better memorization with larger vocab size. Further authors make multiple hypotheses of what might be the underlying explanation hidden behind the improved memorization. After checking these out they conjecture that the reduced sequence length is likely the major contributor explaining the underlying observation.

**Summary Of The Review:**

I set the score as marginally below the threshold given my concerns above where the main one is that the overall conclusions represents empirical findings without much of a contribution towards specific takeaways or practices which would be handy for the community. Nonetheless, I am open to discuss how this work can be positioned in this conference if there will be strong positive feedback from others.

---

> ### Author Response · Authors · 2021-11-19
> **Authors' reply to  Reviewer saPE - part 1**
>
> We are grateful to Reviewer saPE for their feedback. We have updated our text to address their comments. Below, we reply to some of the individual points that they have raised.
>
> >  there is not much of discussion on what can we do with long sequences when we require memorization?
>
> We believe that experiments in our paper highlight two potential avenues for facilitating memorization: (a) by making sequences shorter (e.g., by applying BPE-like algorithms), or, if changing the data representation is not an option, (b) by increasing the expressiveness of the self-attention (see Appendix A and Figure 5). We added this point in the conclusion.
>
> > Authors provide very high-level paths such as 'Our findings can provide guidance for tuning the learned data structures', but I would be very curious to read more details.
>
> What we had in mind is a setup akin to the one in Kraska et al. (2017, [Section 5]), where a binary classifier or a language model are used to memorize a set of strings such that the model assigns a high score only to the strings from the set and low scores to all other strings. In such a setup, generalization has to be avoided - it will lead to false positive errors. If a model can achieve a good memorization while being sufficiently small, it can be used as a substitute or in addition to a Bloom Filter (as used by Kraska et al.) Next, our observed influence between the memorization capacity and the BPE vocabulary size can unlock useful trade-offs in such a task.
>
> > the construction of model parameterization in probing tasks is written on very high-level without any discussion of why this specific method of casting the model as classifier is chosen and how it may or may not provide necessary information towards research questions about memorization, this is mainly addressing section 3.3).
>
> We believe that the way we cast the models as classifiers is intuitive. Indeed, in the case of the LM, our study boils down to measuring  whether it is capable of predicting the last token in a sequence (following the <eos> token), which is the standard setup for an LM. Equally, for MLM, the ability to memorize a masked token should be characteristic of its memory. We now tried to reflect this in text, given the space limitations. Zooming out, our findings are not solely based on casting those architectures as classifiers, e.g. in our experiments on the training data extraction, we use the most standard language modelling setup, without casting it as a classifier.
>
> > right now it is not even clear enough what are 'target class' in the context of vocabulary tokens? I believe explaining this would improve the presentation of that part a lot.
>
> Thanks for this comment, we tried to update the corresponding part of the text. In S2, we described how the memorization capacity of a classifier can be measured, e.g. by measuring its ability to memorize non-systematic labels. For the (masked) language model, we reduce this classification task to predicting a token in a string, with one value of the token corresponding to the class "-1" and the other value corresponding to the class "+1".
>
> >  In addition, authors mention 'SOTA \ state of the art' several times during the work while they do not much relate to the actual SOTA models.
>
> Thanks for raising this, we clarified/removed from the text to avoid any confusion.
>
> > Page 3, "..we prompt the trained LM by a query followed with the separator token and check whether the trained LM reproduces the correct answer within top-1 or top-5 results returned by beam search (beam size 5)." : This is very specific design choice, why did you choose it? Model conditioned on the q gives the entire distribution on sequence-level and beam search is very sensitive to the beam size? I see an alternative as doing unbiased sampling and computing expected similarity with the answer. Some discussion on that would help to clarify this.
>
> Thanks for the suggestion. We want to highlight that extracting similar sequences is not what we are interested in: extracting something similar to the correct answer does not indicate memorisation under our assumption.
> We experimented with beam search with beam size of 5 as it is a default setting in faiseq and the beam search was an "initial" strategy used by some prior work (Carlini et al, 2021). We expect that results under different extraction methods would qualitatively match since BPE is unlikely to be intrinsically biased towards any of them. To support this view, we ran an experiment where for each prompt we sampled  20 continuations (using standard ancestral sampling, T=1.0) and measured how often the actual answer is among them.
> Just as with beam search, the success rate of this attack grows as BPE vocabulary size is increased. We report this experiment in Appendix C (Figure 6).

---

> > ### Comment · Reviewer_saPE · 2021-11-27
> > **Thanks for this response**
> >
> > I thank authors for their detailed response. After reading the review from WXGB, I agree that an experiment with a different task-level uncertainty of the target sequences (such as NMT) would strengthen the findings of this work a lot.
> >
> > I keep my score as is for now, but I wish to increase it if there will be an update (I saw authors mentioned it is on the way) regarding support / contradictions of main findings using some different tasks as I wrote above.

---

> > > ### Author Response · Authors · 2021-11-29
> > > **Reply to Reviewer saPE  (3)**
> > >
> > > > a different task-level uncertainty of the target sequences (such as NMT) would strengthen the findings of this work a lot.
> > >
> > > Unfortunately, we seem to be unable to update the draft directly. Hence we anonymously share our additional experiment using GoogleDrive: https://drive.google.com/file/d/1gA29SGTGa8hMFUeBLHgPnh8PLPDu44S9/view which we will add in Appendix of the paper. In this pdf, we provide a preliminary study on how BPE vocabulary size affects effectiveness of membership attacks on NMT models. In line with our earlier findings, our new results suggest that membership inference attacks become increasingly successful as BPE vocabulary grows.
> > >
> > > Here is a short summary of the experiment. We based it on an English-Dutch translation task, using the OPUS dataset. We randomly sampled two equally-sized 8M sentence sets as training and hold-out sets and used FLORES-101 dev-set for early stopping. Using English and Dutch BPE vocabularies with 10k, 20k, 40k, and 80k merges we trained Transformer-base seq2seq NMT models following a standard fairseq NMT recipe. Finally, we evaluated AUC of the membership inference attack in a procedure similar to that described in S4.2. As a result, we see that models trained under higher-cardinality vocabularies are universally more susceptible to membership inference attacks. Thus we conclude that our findings also apply to Transformer seq2seq models used in NMT.

---

> ### Author Response · Authors · 2021-11-19
> **Authors' reply to Reviewer saPE - part 2**
>
>
> > Bottom of page 3: you used word setups to refer to both tasks and models (right?) which is a bit confusing.
>
> Indeed: on p3 we used the term "setup" as "task", while at the top of page 4 we used it to denote the architecture variant. We fixed this in an updated version. Thank you!
>
> > Page 5: achieves 100% train accuracy : in cases of MLM, was acc measured only on task class prediction or on other masked tokens as well?
>
> Here we refer to the label prediction. We updated the text.
>
> > Page 5: Hence, in this experiment we used a 1-layer: Could you explain why you chose to reduce the model size? In my understanding the task itself might be too easy so the other way is to alter the underlying task (since you specifically construct it).
>
> Thanks for the comment. We decided to simplify the model so that all three can be reasonably represented  on the same figure. The task is indeed constructed, but it is based on a large natural language dataset (SNLI) which is hard to scale sufficiently. Please note that in the remaining experiments, we have used a 4-layer variant, so our findings are not specific to the 1-layer case.
>
> > Was there an early stopping criteria on loss or acc?
>
> In this experiment we do not measure generalization performance, only whether a model can recover a string from the training dataset, given its prefix. By allowing the models to maximally overfit the training data (given a fixed number of updates/epochs fixed across the models for a fair comparison), we find an upper-bound on what it can memorize at all. Thus, we do not use any early stopping in this experiment. We clarified this in the revision.
>
> > In your proposed construction the frequency distribution of tokens doesn't change relatively to each other, right?. Is it the same for BPE learning when number of merges is growing? Would be great to show this....
>
> Thank you a lot for this comment! A similar point was also raised by Reviewer WXGB. In Appendix D, we have added a study that investigates the changes in relative frequencies of the tokens when growing the BPE vocabulary size and whether it can cause the changes in the memorization capabilities. Below is a summary of this study.
>
> We start by showing that, indeed, the size of the BPE-learned vocabulary affects the relative distribution of the tokens, with larger vocabulary size making it increasingly peaky. Next, we take a dataset of randomly generated strings (same as in the main text S5.1) that has uniform distribution of tokens (0s and 1s) and repeatedly apply an "idealized" subword learning procedure that groups all possible pairs of tokens. At each step, this procedure maintains the uniform distribution of tokens, shortens the string length by a factor of two and grows the vocabulary size quadratically. Using the obtained sequence of dataset "views" we repeat the random label memorization experiments and confirm that, still,memorization becomes increasingly successful as string length (measured in tokens) decreases. Thus we can also rule out the possibility that the observed effect is due to changes in the relative frequencies of tokens - as it persists when there is no change.

---

### Official Review · Reviewer_KAZC · 2021-11-02

**Correctness:** 3
**Technical Novelty And Significance:** 3
**Empirical Novelty And Significance:** 3
**Recommendation:** 6
**Confidence:** 4

**Main Review:**

I think this paper presents a thorough evaluation on how bep affects transformer’s memorization. My main concern is that their conclusion, shorter input sequences lead to better memorization, is drawn a bit hastily. The paper draws this conclusion indirectly by disproving the other two assumptions that they gave. However, are the 3 assumptions the only possible candidate theories? Or, are there ways to directly prove that it is the sequence length that makes a difference? One possible experiment can be, in the experiment of training data recovery, you train two different language models with the same data set, while for one you cut the long  input sentences into multiple short sequences (or concatenate sentences to make the input sequence longer), for the other you train the language model with the original data set.

**Summary Of The Paper:**

This paper as the name suggests, tries to figure out whether and through which ways bpe can affect Transformer’s memorization capacity. It evaluates Transformer’s memorization under these 4 settings: memorizing random synthetic data, memorizing random labeled natural language data, recognizing training data with lower output entropy, and training data recovery QA. Experiments show that if we have more merge times for bpe (larger vocabulary size), the model’s performance on all the four settings would improve on 3 architectures, which shows that bpe indeed can affect the model’s memorization capacity. Larger the vocabulary size is, the better the model would perform on memorizing. Then, the paper tries to figure out why more merges in bpe would lead to better memorization. By excluding two other hypotheses, the paper concludes that more merges in bpe results in shorter input sequence and that makes memorizing easier for transformers.

**Summary Of The Review:**

Thorough empirical study on whether BPE affects memorization in transformers. While the final conclusion, shorter input sequences lead to better memorization, is not well supported by experiments.

---

> ### Author Response · Authors · 2021-11-16
> **Reply to Reviewer-KAZC**
>
> Thank you for your review and comments!
>
> We find your suggestion very interesting, however we would like to kindly ask for more details on how to implement the proposed experiment right. Indeed, if we understood it correctly, concatenating or splitting sequences would lead to altering the underlying dataset (e.g. having more individual sequences in the case of splitting). This would result in new confounding effects which would need to be controlled too.

---

> ### Author Response · Authors · 2021-11-30
> **Reply to Reviewer KAZC (2)**
>
> Maybe you will find it interesting - reviewers WXGB & saPE suggested to demonstrate that our findings also generalize to NMT scenario and we have run a corresponding experiment. In this pdf  https://drive.google.com/file/d/1gA29SGTGa8hMFUeBLHgPnh8PLPDu44S9/view we demonstrate that our findings can transfer to NMT, too: we show that larger BPE vocabulary size increases susceptibility of Transformer NMT models to membership inference attacks. We will include this experiment into Appendix.

---

### Decision · Program_Chairs · 2022-01-20

**Decision:**

Reject

**Comment:**

This paper investigates the role of BPE and vocabulary sizes in memorization in transformer models. Through a series of experiments on random label prediction, training data recovery and membership inference attacks, the paper shows that larger vocabulary sizes lead to improved memorization. The Reviewers all agree that the paper investigates an important question and does so thoroughly. The main concerns were about: (1) the validity of the conclusion that it is sequence length indeed which affects memorization; and (2) the lack of more tasks to validate the findings. For (1) the authors added another set of experiments which further rule out frequency effects as a factor, but I agree with Reviewer KAZC that more evidence is needed which directly shows that sequence length is responsible (e.g. are shorter PAQ questions memorized better?). For (2) the authors shared a google drive link with additional results on NMT after the deadline, which the reviewers appreciated. Overall, however, the paper needs more work in order to unify all these results in a single draft.